



# Microphysical characterization of long-range transported biomass burning particles from North America at three EARLINET stations

Pablo Ortiz-Amezcua[1,2], Juan Luis Guerrero-Rascado[1,2], María José Granados-Muñoz[1,3], José Antonio Benavent-Oltra[1,2], Christine Böckmann[4], Stefanos Samaras[4], Iwona S. Stachlewska[5], Łucja Janicka[5], Holger Baars[6], Stephanie Bohlmann[6] and Lucas Alados-Arboledas[1,2]

[1]Andalusian Institute for Earth System Research (IISTA-CEAMA), 18006, Spain
[2]Department of Applied Physics, University of Granada, 18071, Granada, Spain
[3]NASA/JPL/California Institute of Technology, Wrightwood, CA, USA
[4]Institute of Mathematics, Potsdam University, 14469 Potsdam, Germany
[5]Institute of Geophysics, Faculty of Physics, University of Warsaw (IGFUW), 02-093 Warsaw, Poland
[6]Leibniz Institute for Tropospheric Research, 04318 Leipzig, Germany

*Correspondence to*: Pablo Ortiz-Amezcua (portizamezcua@ugr.es)

**Abstract.** Strong events of long-range transported biomass burning aerosol were detected during July 2013 at three EARLINET stations, namely Granada (Spain), Leipzig (Germany) and Warsaw (Poland). Satellite observations from MODIS and CALIOP instruments, as well as modeling tools such as HYSPLIT and NAAPS have been used to estimate the sources and transport paths of those North American forest fire smoke particles. Multiwavelength Raman lidar technique was applied to obtain vertically-resolved particle optical properties, and further inversion of those properties with regularization algorithm allowed for retrieving microphysical information on the studied particles. The results highlight the presence of smoke layers of 1-2 km thickness, located at about 5 km asl altitude over Granada and Leipzig, and around 2.5 km asl at Warsaw. These layers were intense, as they accounted for more than 30 % of the total AOD in all cases, and presented optical and microphysical features typical for different aging degrees: color ratio of lidar ratios ($LR_{532}/LR_{355}$) around 2, α-related Angström exponents of less than 1, effective radii of 0.3 μm, and large values of single scattering albedos, nearly spectrally independent. The intensive microphysical properties were compared with columnar retrievals form co-located AERONET stations. The intensity of the layers was also characterized in terms of particle volume concentration, and then an experimental relationship between this magnitude and the particle extinction coefficient was established.

## 1 Introduction

Atmospheric aerosols are known to play an important role on effective radiative forcing because of their interactions with radiation and clouds (Boucher et al., 2013). These interactions are strongly dependent on scattering and absorption capabilities of the aerosol particles, and on vertical distribution of the aerosol plumes along the atmospheric column. In particular, biomass burning particles can have completely opposed behavior, depending on their content in organic and black carbon, on their size and on their spatial distribution in the atmosphere. These properties of the biomass burning aerosol particles are affected by source type, combustion type and phase (Martins



et al., 1998; Jacobson, 2001; Reid et al., 2005 a, b), and so-called aging process caused by different mechanisms such as photochemical oxidation (Grieshop et al., 2009 a, b), hygroscopic growth (Hobbs et al., 1997; Granados-Muñoz et al., 2015; Titos et al., 2014 a, b, 2016) or coagulation (Fiebig et al., 2003).

5    It has been demonstrated that large smoke plumes from large forest fires can be injected into the free troposphere, and then easily transported by air masses along the Earth, presenting long residence times in the atmosphere (Andreae, 1991; Fromm et al., 2000; Fromm and Servranckx, 2003; Jost et al., 2007; Rosenfeld et al., 2007; Freitas et al., 2007; Peterson et al., 2014; Seinfeld and Pandis, 2016; Guerrero-Rascado et al., 2010, 2011). The study of these aerosol transport 10    processes is relevant for all aerosol types, since this information is crucial in modeling the global impact of aerosol particles and monitoring events of social relevance (Pappalardo et al., 2013).

In this sense, global and continental networks are necessary, as they can provide appropriate spatial distribution of measurements with enough quality to fairly account both for the impact of isolated events as for the climatological effect of atmospheric aerosol particles (as opposed to satellite, with 15    much higher spatial resolution but less likely to be equipped with instruments with the same potential and complexity as the ground-based systems). EARLINET, the European Aerosol Research Lidar Network (Pappalardo et al., 2014) is an established network with the main goal of providing a database for distribution and properties of the aerosol over Europe, exhaustive and complete enough to be climatologically significant. Thanks to the use of lidar technique as the basis 20    of the network, information on vertical distribution of atmospheric aerosol particles with large spatial and temporal resolution is provided.

In this work, intense events of biomass burning particles released from North American forest fires during summer 2013 are analyzed in terms of particle microphysical properties when they reached different EARLINET stations after being transported by air masses across the Atlantic Ocean. 25    Summer 2013 was one of the driest in the previous decades in USA and Canada. Large forest fires caused by thunderstorms started at the end of June 2013 and continued being active during July and August, causing vast forest areas to burn up (Ancellet et al., 2016). In a previous work (Ortiz-Amezcua et al., 2014), a preliminary optical description was given for the lidar detection of a smoke event over Granada (Spain) in July 2013. Markowicz et al. (2016) used in-situ 30    measurements, passive and active remote sensing observations, as well as numerical simulations to describe the temporal variability of aerosol optical properties for the same period over Poland, and Janicka et al. (2016) studied the properties of the mixing of those smoke particles with dust particles over Warsaw. Ancellet et al. (2016) reported optical properties of the smoke plumes transported over some stations in the Western Mediterranean Basin in June 2013. Veselovskii et al. 35    (2015) described vertically-resolved optical and microphysical properties of particles detected in Washington, DC coming from similar forest fires after regional transport in August 2013.

Given the importance of smoke transport events, several attempts have been made at establishing mean values and ranges for the reported main optical and microphysical properties of the biomass





burning particles, classifying them according to source regions, combustion phase and aging (Dubovik et al., 2002; Reid et al., 2005 a, b; Müller et al., 2007a). These estimations are strongly dependent on the detection type (in-situ measurements, passive or active remote sensing), and every new measurement can show a different feature which does not fit with those reported in the

mentioned works. This paper intends to make a significant contribution to the general knowledge about biomass burning events detected after transatlantic transport, not only giving new observed values of intensive properties of the particles, but highlighting the similarities and differences among presumptive different events. These concordances or discrepancies will be meaningful, taking into account that the three analyzed plumes are different in terms of origin, transport path

and conditions at each observation site, and they will allow for extracting some common pattern for transatlantic transport to be applied in future events.

We present a complete microphysical characterization of the smoke particles released into the free troposphere during different North American forest fires at the beginning of July 2013 and detected 8-10 days after, over three EARLINET stations (Granada, Leipzig and Warsaw) at different times

and altitudes. Vertically-resolved microphysical properties after such long-range transport are necessary in order to account for the particle properties that might have changed during the process and that might then directly affect their radiative impact. Raman lidar allows for stand-alone (nighttime) microphysical retrievals, i. e., the calculation of particle microphysical properties using just $3\beta + 2\alpha$ set of lidar optical variables: particle backscatter coefficients at three wavelengths

(355, 532 and 1064 nm) and particle extinction coefficients at two (355 and 532 nm), (Müller et al., 1999; Böckmann, 2001).

## 2. Experimental sites and instrumentation

Three European experimental sites were selected in this work for characterizing the detected smoke plumes (Table 1). These stations are part of EARLINET network and have lidar systems that fit the

conditions for obtaining particle microphysical properties using regularization algorithms. That is, the so-called "$3\beta + 2\alpha$" optical data set can be obtained, since the three lidar systems are able to detect elastic signals at the emitted wavelengths 355, 532 and 1064 nm, and $N_2$ Raman-shifted signals at 387 and 607 nm.

Moreover, columnar microphysical data from sun-photometers at three AERONET (Holben et al.,

1998) stations have been used. The sites were selected to be the nearest AERONET stations to the EARLINET stations GR, LE and WA. For Granada, where two photometers from the network were working during the studied period, the one located on the hill "Cerro de los Poyos" (37.11° N, 3.49° W, 1830 m a.s.l.) was selected because it presents the advantage of being more than 1 km higher than the lidar station, making easier to study aerosol layers decoupled from PBL (Granados-

Muñoz et al., 2014). In Leipzig, the selected photometer was co-located with the lidar system. In the case of Warsaw, the nearest AERONET station was found at the Geophysical Observatory at Belsk (51.84° N, 20.79° E, 190.0 m a.s.l.).





Cerro de los Poyos is around 12 km apart from Granada, and the observatory at Belsk is located at a distance of about 50 km South of Warsaw. Although these distances can be considered negligible as compared to the much larger horizontal scale of the common air masses (Holton, 1992), special care was taken when comparing the results from Raman lidar and from sun-photometer techniques.

**3. Methodology**

In the first part of this work, satellite observations and models were used to study the sources and transport mechanism of the detected aerosol particles.

The Active Fire Mapping Program (http://activefiremaps.fs.fed.us/), a satellite-based fire detection and monitoring program managed by the USDA Forest Service Remote Sensing Applications
Center (RSAC) was used to analyze the distribution of fires in the United States and Canada during the studied period. High temporal image data collected by the NASA's Moderate Resolution Imaging Spectroradiometer (MODIS) on Terra and Aqua platforms are currently the primary remote sensing data source of this fire detection program. MODIS provides multiple daily observations of the United States and Canada, which is ideal for continuous operational monitoring
and characterization of wildland fire activity.

NAAPS (Navy Aerosol Analysis and Prediction System) model of Marine Meteorology Division, Naval Research Laboratory (NRL), (http://www.nrlmry.navy.mil/aerosol/) was used for forecasting aerosol optical depth and particle density of smoke at the Earth's surface.

The analysis of backward trajectories was performed by means of the HYSPLIT model (Hybrid
Single-Particle Lagrangian Integrated Trajectory) (Stein et al., 2015; Rolph, 2016) developed by the NOAA (National Oceanographic and Atmospheric Administration) in collaboration with the Australia's Bureau of Meteorology. Two types of multiple trajectory analyses were carried out: cluster analysis and ensemble calculation. For the illustration of airflow patterns in order to interpret the transport over different spatial and temporal ranges, trajectories that have some
commonalities in space and time were merged into groups, called clusters, and represented by their mean trajectory. Differences between trajectories within a cluster were minimized while differences between clusters were maximized (Draxler et al., 2009). The ensemble form of the model (instead of single trajectory calculation) was used to trace back the history of the detected layers with the objective of improving plume simulations and accounting for possible uncertainties. With this
method, multiple trajectories start from the selected starting point, and each member of the trajectory ensemble is calculated by offsetting the meteorological data by a fixed grid factor, resulting in 27 members for all-possible offsets in longitude, latitude and altitude.

The observations of the spaceborne CALIOP (Cloud-Aerosol Lidar with Orthogonal Polarization) were used to track the aerosol plumes during their transport. This lidar system, with two
wavelengths (532 and 1064 nm), polarization channels at 532 nm, an infrared radiation radiometer and a Wide Field Camera, is on board CALIPSO (Cloud-Aerosol Lidar Infrared Pathfinder Satellite



Observations) mission, launched in 2006. Its main products are attenuated backscatter profiles and also clouds and aerosol information together with layer properties (Winker et al., 2009).

In the second part of the work, vertical profiles of optical properties (independently retrieved particle backscatter and extinction coefficients, Angström exponents and lidar ratios) were obtained from night-time lidar measurements applying the Raman methodology (Ansmann et al., 1992). The uncertainties in the optical properties were determined by means of a numerical procedure based on the Monte Carlo technique, commonly used in the EARLINET network (Guerrero-Rascado et al., 2008; Pappalardo et al., 2004; Mattis et al., 2016).

The set of 3β + 2α obtained from Raman lidar observations was employed to obtain particle microphysical properties (i. e., particle volume concentration, effective radius, complex refractive index and single scattering albedo) using an inversion algorithm developed at the University of Potsdam, UP (Böckmann, 2001; Böckmann et al., 2005). This method has been developed in the framework of EARLINET (Müller et al., 2016) and is based on explicitly solving the mathematical equations that relate the particle microphysical and optical properties by means of regularization techniques, an approach that is shared with Müller et al. (1999) and Veselovskii et al. (2002) inversion algorithms. That means that forward computations using tables containing microphysical versus optical properties are not carried out, having the advantage that particle size distribution shape is not assumed as input, but approximately calculated as output. A detailed description of the approach and software of UP algorithm was published by Böckmann (2001), Böckmann et al. (2005), Osterloh et al. (2009, 2011, 2013) and Samaras et al. (2015).

## 4. Results

### 4.1 Characterization of sources and transport of the smoke plumes

According to MODIS fire detection maps (Fig. 1, left), several active forest fires were detected at the United States and Canada, releasing large amounts of biomass burning particles during July 2013. Figure 1 (right) shows the smoke surface concentration at the beginning of that month, given by NAAPS model. High concentrations can be observed in almost all North America, reaching values more than 64 µg/m3 in several regions. Markowicz et al. (2016) studied the relative aerosol optical depth (AOD) changes in several North American AERONET stations during the first weeks of July 2013, finding values reaching 1.5 (at 500 nm), which implies mean AOD anomalies (with respect to long-term means for July) up to 0.42.





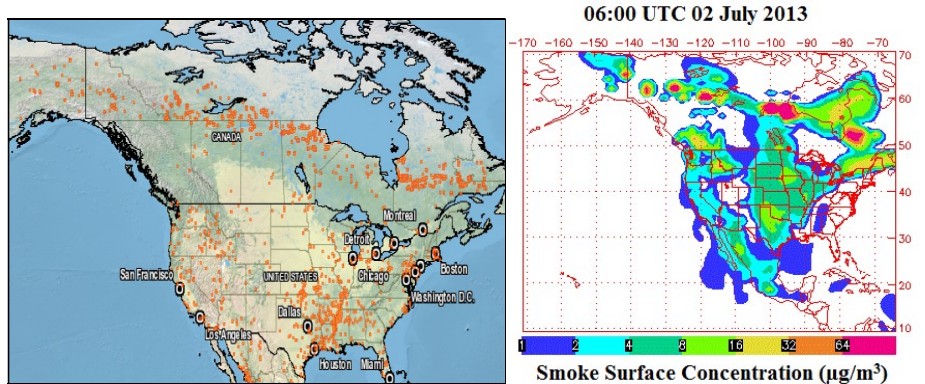

**Figure 1:** Active Fire map for the period from 1st July to 15th July 2013 (left) and concentration of smoke particles at the surface according to NAAPS model, for a specific time (2nd July 2016 at 06:00 UTC) within the period of intense forest fires in North America (right).

The cluster analysis performed using Hysplit software revealed that, during June and July 2013, the prevailing synoptic situation favored the transport of these aerosol particle plumes across the Atlantic Ocean to Europe. In Fig. 2, the most relevant 10-days backward trajectories clusters for each of the studied stations and layers are represented. This figure shows the main influence of air masses coming from North America, accounting for 59% of all the trajectories ending at Granada,
64% for Leipzig and 61% for Warsaw.

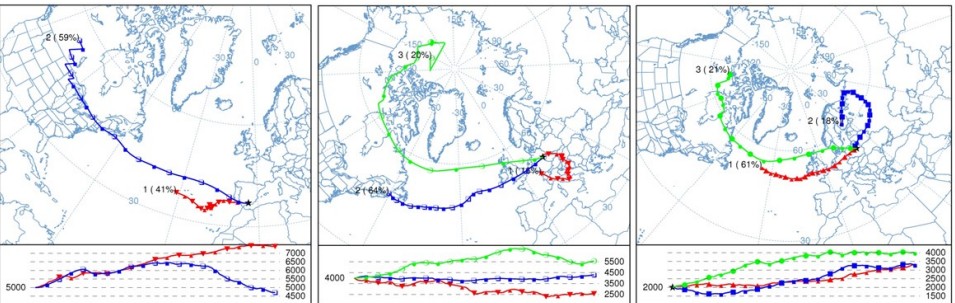

**Figure 2:** Coordinates and altitude in meters above ground level (black lines) of the 10-day backward-trajectories clusters during the period June-July 2013 arriving at Granada (left), Leipzig (center) and Warsaw
(right).

     Using EARLINET database, it was confirmed the detection of possible smoke particles for the three selected stations. In Figure 3, the lidar raw corrected signal in the selected locations shows the presence of aerosol layers at different altitudes. In Granada and Leipzig, multilayer structures were found and smoke particles appeared in the free troposphere, between 4 and 6 km above sea level
(a.s.l.), while in Warsaw, the high load of aerosol particles was observed at a lofted aerosol layer between 1.5-3 km a.s.l. This layer was decoupled from the aerosol layer near to the surface, as it can be seen in Fig. 3.





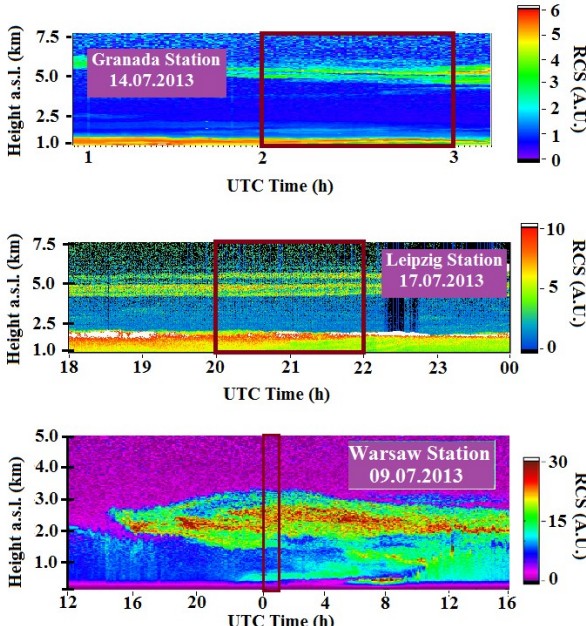

**Figure 3:** Time evolution of lidar raw corrected signal (in arbitrary units) at 1064 nm showing the detection of the smoke plumes at Granada, Leipzig and Warsaw stations, with analyzed intervals inside red boxes.

Ensembles of backward trajectories generated with HYSPLIT model were used to determine the origin of the air masses carrying aerosol plumes arriving at the studied stations at the relevant heights (Fig. 4). They confirmed that the relevant air masses came from superficial layers over North American forest fires detected by MODIS, and that they were advected for around 8-10 days before reaching the stations.

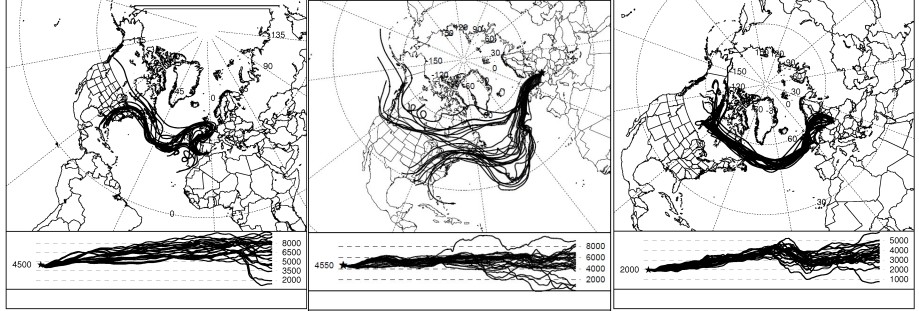

**Figure 4:** Coordinates and altitude in meters above ground level (black lines) of the ensembles of probable air masses trajectories ending at the investigated layer on 14[th] July 2013 above Granada (left), 17[th] July 2013 above Leipzig (center) and 9[th] July 2013 avobe Warsaw (right).

These ensemble trajectories in Fig. 4 also show that in contrast to the aforementioned general transport from North America (as seen in cluster analysis, Fig. 2), there are two clearly different source zones for the specific analyzed layers. While the layer arriving to Warsaw unequivocally



comes from West Canada, the corresponding layers arriving to Granada and Leipzig might come both from West Canada and from East USA. This difference in source region implies different types of forest: coniferous forests predominate in Canada while in that part of USA, deciduous forests are the most important (David and Holmgren, 2001). This might be crucial, since it implies

5  different fuel and combustion type (modifying the black carbon content) and thus have to be taken into account when analyzing the physical properties of the detected particles.

The geolocation of CALIPSO overpasses and backward trajectories on Fig. 4 provide a reliable tool to assess the involvement of those air masses in the transportation of the smoke plumes which finally reached Europe. Figure 5 illustrates some of the overpasses of this satellite coinciding in

10  space and time with parts of the back-trajectories on 1st-8th July 2013 for Warsaw case and 5th-16th July for Granada and Leipzig cases. The aerosol type product (Omar et al., 2009) provided by CALIPSO (Figures 6 and 7) confirmed that the smoke columns reached 5 km altitude over the sources, and the smoke content on the transported air plumes, as indicated by the black color.

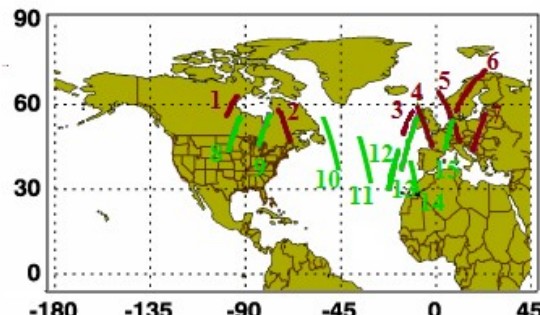

15  **Figure 5:** Map showing the relevant CALIPSO overpasses tracking some smoke plumes being transported to Europe. Brown lines stand for plumes arriving at Warsaw on 9th July 2013, and Green lines stand for plumes arriving at Granada on 14th July 2013 and Leipzig on 17th July 2013.



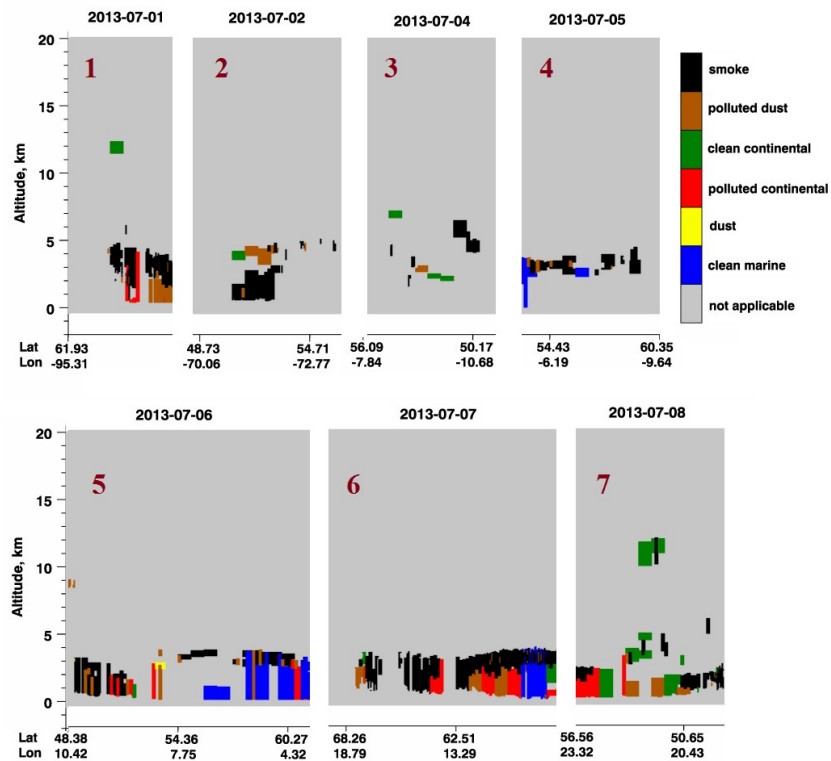

**Figure 6:** Position (altitude, latitude and longitude) and type of the aerosol layers detected by CALIPSO for each of the overpasses tracking the masses arriving at Warsaw (depicted in Fig. 5 as 1-7), black color indicating smoke aerosol particles.





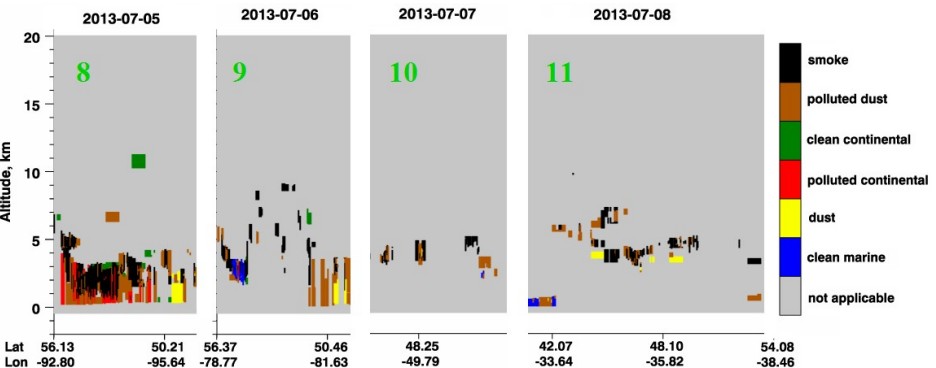

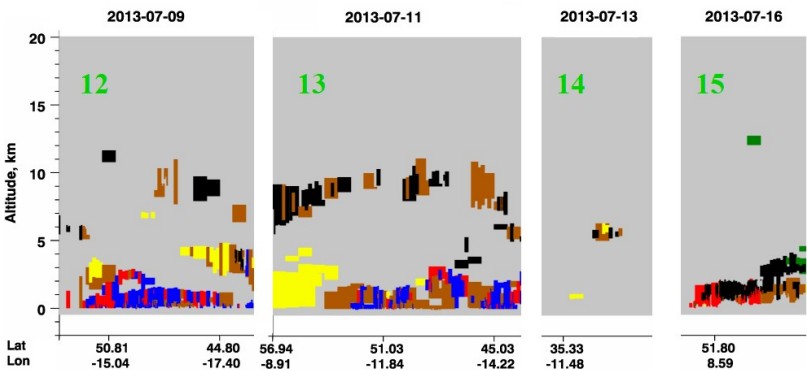

**Figure 7:** Position (altitude, latitude and longitude) and type of the aerosol layers detected by CALIPSO for each of the overpasses tracking the masses arriving at Granada and Leipzig (depicted in Fig. 5 as 8-15), black color indicating smoke aerosol particles.

## 4.2 Optical properties

In order to characterize the optical properties of the biomass burning particles, the Raman algorithm was applied to lidar data corresponding to 2-3 h (UTC) on 14[th] July for Granada station, 20-22 h (UTC) on 17[th] July for Leipzig and 0-1 h (UTC) on 9[th] July for Warsaw. Particle backscatter coefficient ($\beta$), particle extinction coefficient ($\alpha$), lidar ratio (LR) and linear particle depolarization ratio ($\delta_P$) profiles are plotted in Fig. 8. The regions of profiles affected by incomplete overlap and by too low backscatter ratio are not shown. The $\beta$- and $\alpha$-profiles highlighted that the smoke layers were intense in terms of optical properties, and the low $\delta_P$ values (less than 4% for Granada and Warsaw, less than 8% for Leipzig) indicate the large proportion of spherical, fine-mode particles (Navas-Guzmán et al., 2013; Granados-Muñoz et al., 2014; Bravo-Aranda et al., 2015).

The thickness of each smoke layer was calculated using the gradient method (Flamant et al., 1997), and it was obtained that the bottom and top of each layer was 4.3-6.1 km at GR, 4.2-5.7 km at LE and 1.5-3.3 km at WA. By integration of the particle extinction coefficient over the smoke layer,



the fraction of the total AOD associated to the smoke plume was derived, obtaining that it accounted for more than 40% of the total AOD (532 nm) in the case of Granada, more than 30% in Leipzig, and more than 70% in Warsaw. In these calculations, the extinction coefficients along the region of incomplete overlap were approximated by multipliying the backscatter coefficient profile at this region (which is not affected by incomplete overlap) by a constant LR.

In each case, a single thin layer (200 m thick for GR and WA, and 300m for LE) was selected (pointed with brown rectangles in Fig. 8) to obtain an optical and microphysical description of the transported particles. In Table 2, the main optical properties of the analyzed aerosol layers are shown. Very similar properties were found for Granada and Leipzig, with low extinction-related Angström Exponents (AEα) and LR of 23 ± 10 and 25 ± 4 sr for 355 nm, and 51 ± 11 sr and 51 ± 9 sr for 532 nm. The very low measured LR values at 355 nm represent a feature to point out, since they indicate low absorption from the smoke particles, compared to the mean value of 46 ± 13 sr for North American biomass burning particles reported by Müller et al. (2007a). However, Müller et al. (2005) already found LR355 ranging from 21 to 67 sr for biomass burning aerosol, which agrees with the values here presented. The "color ratio of lidar ratios" ($CR_{LR}$ = LR532/LR355) reached values around 2 for GR and LE, which hints towards the aging process. It has been demonstrated that $CR_{LR} < 1$ is usual for fresh smoke particles, while $CR_{LR} > 1$ corresponds to aged smoke (Müller et al., 2005; Müller et al., 2007a; Alados-Arboledas et al., 2011; Nicolae et al., 2013). The latter comparison among the results obtained and other values found in the literature about biomass burning events detected in Europe is summarized in Table 3.

Concerning the values obtained for Warsaw, there are noticeable differences with the other two stations: higher $AE_\alpha$ (reaching 0.98 ± 0.06) and LR, and slightly lower $CR_{LR}$ (although it keeps well over 1, being consistent with the aging during the transport). These discrepancies might be due to the different smoke sources as observed in section 4.1, but may also be attributed to a different aging process.





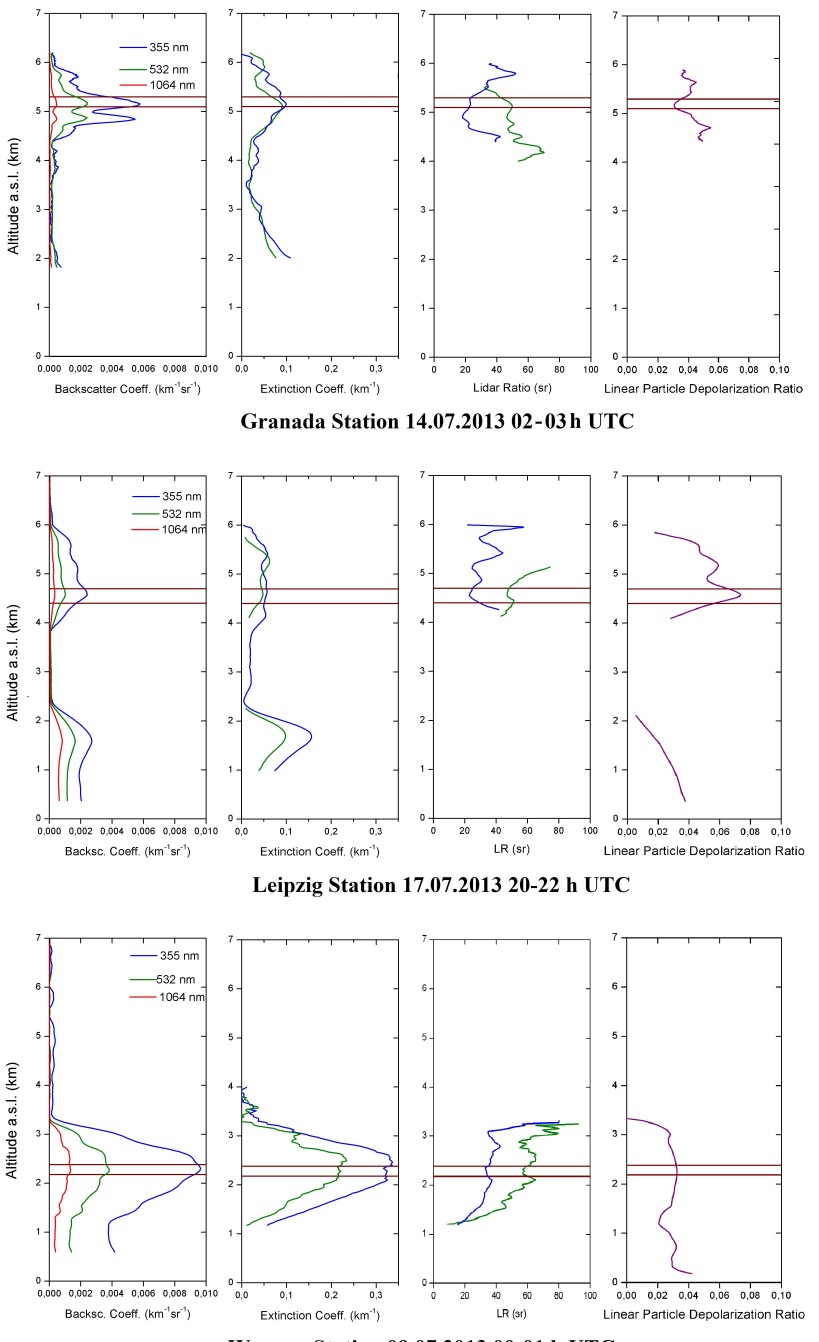

**Granada Station 14.07.2013 02‑03 h UTC**

**Leipzig Station 17.07.2013 20-22 h UTC**

**Warsaw Station 09.07.2013 00-01 h UTC**

**Figure 8:** Vertical profiles of particle backscatter and extinction coefficients, lidar ratio and linear particle depolarization ratio obtained for Ganada, Leipzig and Warsaw cases. The layers analyzed to obtain the optical and microphysical description of the transported particles are marked with brown rectangles.





### 4.3 Microphysical properties

The UP inversion algorithm was applied to the selected layers in Table 2 in order to retrieve a microphysical description of the detected aerosol particles. Table 4 shows the results obtained.

The retrieved particle volume concentrations ($C_v$) present values over 10 $\mu m^3 cm^{-3}$, reaching almost 35 $\mu m^3 cm^{-3}$ in Warsaw. The effective radii present high values in agreement with the aging process, and fit the exponential curve derived by Müller et al. (2007b) with a discrepancy below 15 % for Granada and Leipzig, and 20 % for Warsaw. Complex refractive indices have real part (RRI) a bit lower than 1.50, which represents the typical value for Boreal Forest Fires particles according Dubovik et al. (2002), see Table 3.

Very low imaginary part of the refractive index (IRI) with values form 0.0012 to 0.003 compared to 0.0094 ± 0.003 given by Dubovik et al. (2002) and single scattering albedos (SSA) close to 1 indicate a weak absorption by the particles, and therefore a low black carbon fraction, in disagreement with some previous works about biomass burning particles (Wandinger et al., 2002; Alados-Arboledas et al., 2011) but in agreement with others (Eck et al., 2009; Samaras et al., 2015). The spectral dependence of the SSA between 355 nm and 532 nm shows what could be considered an anomalous behavior compared to some columnar retrievals (Reid et al., 2005 a, b; Dubovik et al., 2002), where biomass burning aerosols SSA typically decreases with increasing measurement wavelength. However, the nearly constant or slightly positive spectral dependence is also found in other studies (Eck et al., 2009; Alados-Arboledas et al., 2011; Pereira et al., 2014). It is noteworthy, that the refractive index is assumed wavelength-constant for the inversion algorithm used in this work, and thus the size distribution plays a major role in SSA retrieval. In the studied cases, it is found that the fine modes of the retrieved size distributions are broad (Fig. 9), which implies a contribution of larger particles that cancels out the typically negative spectral dependence of SSA. The different spectral behaviors and ranges of the SSA in the mentioned works are not only related to aging process, because similar properties have been found for both fresh and aged biomass burning particles. These properties depend also on burning region and on fuel and combustion type.

An important feature of these results is the similar intensive properties found for particles detected in Granada and Leipzig, as compared to those retrieved for Warsaw. Such similarities and differences are consistent with the optical properties and they are attributed to the different source region of the smoke plumes, as explained in section 4.1. Additionally, the pathways of the plumes arriving to GR and LE did coincide until a certain point, as also shown in that section.



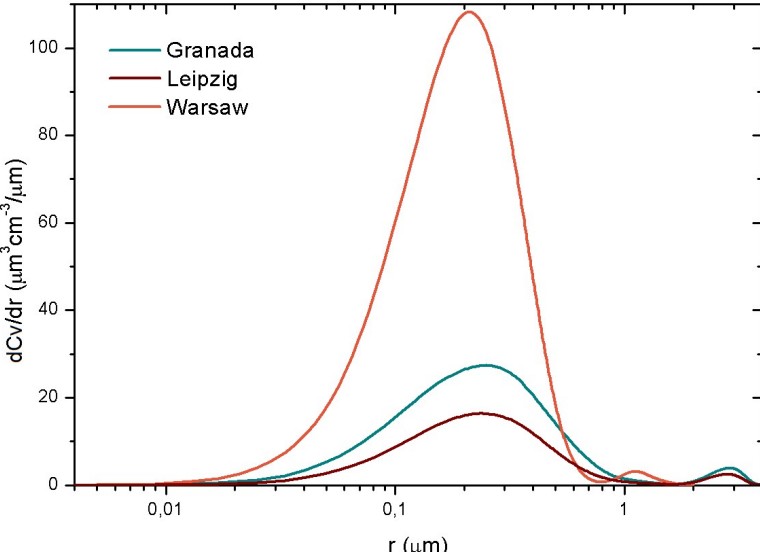

**Figure 9:** Volume particle size distributions retrieved for the selected points of the analyzed smoke plumes.

The integrated volume concentration along the smoke layers ($V_L$) was also calculated in order to make a comparison with AERONET retrievals of integrated volume concentrations along the whole atmospheric column (V). A triangle-shape approximation was used for the $C_v$ profile of the 1.8 km thick layer for Granada and Warsaw (at 4.3-6.1 and 1.5-3.3 km a.s.l., respectively), taking $C_v = 0$ at the points corresponding to the smoke layer top and the smoke layer bottom, and the calculated $C_v$ values reported in Table 3 for the selected altitude. The integrated concentration for the case of Leipzig was approximated using a rectangle-shape $C_v$ profile of the 1.5 km thick layer (at 4.2-5.7 km a.s.l.). These approximations may be justified by looking at the shape of the particle extinction profiles in Fig. 8. Fine and coarse modes distinction ($V_L^f$ and $V_L^c$, respectively) was also calculated, using the same inflection points as given by AERONET.

Table 5 shows the found values, which highlight again that the plume observed over Warsaw was more intense (the $V_L$ at this station doubles the values at the two other stations) and also that fine-mode particles were the most important ones. This mode represents 69% of total $V_L$ at GR, 63% at LE and 95% of total $V_L$ at WA.

Once the integrated concentration of each layer was calculated, an assessment of their impact on the total atmospheric column was made. Three AERONET microphysical retrievals were then selected, using the stations mentioned in section 2. For each station, it was selected the closest in time AERONET retrieval that according to the columnar AERONET retrieved properties showed a clear presence of the detected smoke plume. The times were 06:29 UTC for Cerro Poyos (Granada), 17:31 UTC for Leipzig and 04:23 UTC for Belsk (Warsaw), corresponding to 3:30 h after lidar





measurements at GR, 2:30 h before lidar measurements at LE and 3:20 h after lidar measurements at WA.

Table 6 shows the volume concentration in the whole atmospheric column (V) provided by AERONET, distinguishing among total, fine and coarse modes. It is seen that the fine mode fraction is high in all cases, as it was observed for $V_L$ in Table 4. It can be also seen that the smoke layer detected at GR during the night presented $V_L$ that represents 43% of the total V observed during the afternoon; $V_L$ at LE was 22% of the V observed during the day; and 57% of the V during day at Belsk was observed for the smoke layer over WA.

The main intensive microphysical properties retrieved from AERONET algorithm are also included in Table 6. The low absorption of the analyzed particles is confirmed, with very low IRI and very high SSA. The IRI included are the average values over all the wavelengths retrieved from photometers. The wavelengths at which SSA were obtained are different from lidar wavelengths, thus ultraviolet (UV) and visible (VIS) ranges are compared. SSA values appear to be almost spectrally independent, but slightly decreasing with wavelength. This slope does not agree with the lidar retrieval presented previously in this section, but does agree with other studies using only columnar retrievals (Reid et al., 2005 a, b; Dubovik et al., 2002).

Concerning the effective radii, discrepancies with values from Table 4 around 20%, 30% and 10% are found for GR, LE and WA, respectively. These differences are small taking into account the spatial and temporal differences among the measurements, and also the volume investigated. Real refractive indices (RRI) are also around 1.5 for photometric retrievals, although in LE a RRI of 1.43 was found.

The experimental relationship between particle volume concentration and particle extinction coefficient at 532 nm is also analyzed in this study. In addition to the three cases illustrated in Fig. 8 and Tables 2 and 4, three more cases for the same day in Granada, one more case for 11[th] July 2013 in Leipzig, and four more cases for 8-10[th] July 2013 in Warsaw were calculated with UP algorithm. The points from the additional cases along with some points from other cases in literature (Veselovski et al., 2015; Janicka et al., 2016) were plotted, see Fig. 10. It was found that a linear dependence can be deduced. A linear fit using $\alpha_{532}$ was calculated, obtaining $C_v(\mu m^3 cm^{-3}) = (3 \pm 1)(\mu m^3 cm^{-3}) + \alpha_{532}(0.130 \pm 0.006)(\mu m)$ with $R^2 = 0.95$. The resulting linear parameters can be thus assumed to be representative for the approximation of volume concentration values in events of biomass burning particles transported from North America to Europe when $\alpha_{532}$ is available. Nevertheless, this linear parametrization should only be applied for aerosol particles with similar chemical composition and affected by similar ageing processes as the ones presented here due to the large dependence of the aerosol properties on these factors.





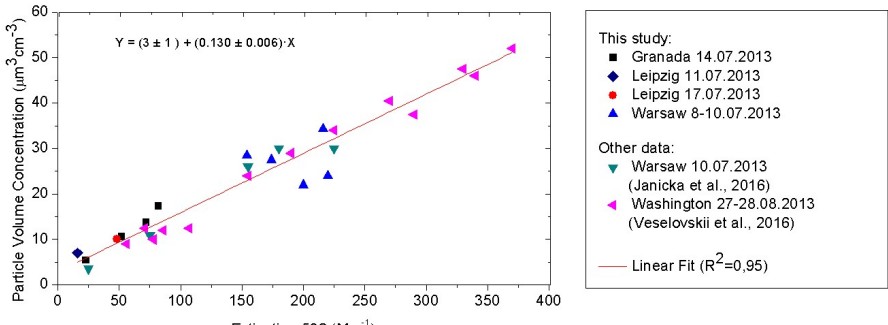

**Figure 10:** Scatter plot of the particle volume concentration as a function of the extinction coefficient at 532 nm. The red line represents the linear regression.

## 4. Summary and conclusions

In the present work, a complete optical and microphysical analysis of biomass burning aerosol particles transported from North American forest fires to Europe was presented. The event occurred during July 2013, and Raman lidar data from three EARLINET stations (Granada, Leipzig and Warsaw) were used in order to obtain independent particle backscatter and extinction coefficient profiles and thus to apply a regularization inversion method developed at the University of Potsdam to retrieve particle microphysical properties.

The observed smoke layers, with thickness between 1 and 2 km, presented AOD (532 nm) that accounted for more than 40 % of total AOD at GR, more than 30 % of total AOD at LE, and more than 70 % of total AOD at WA. Lidar ratios in the range 23-34 sr for 355 nm, and 51-58 sr for 532 nm were obtained, what means color ratio of lidar ratios around 2. These values, together with α-related Angström exponents ranging 0.20-0.98, are in agreement with other studies about biomass burning particles aging process due to transport, although a minor effect was found for Warsaw case.

UP inversion algorithm was applied to optical profiles from Raman lidar data to obtain the microphysical properties of the detected particles. Particle volume concentrations of $17.3 \pm 0.2$, $10.1 \pm 0.4$ and $34.3 \pm 0.7$ $\mu m^3 cm^{-3}$ were found for the layer peaks at Granada, Leipzig and Warsaw, respectively. Effective radii between 0.207 and 0.34 μm were derived, values that approximately fit an exponential dependence with transport time given in a previous article. Very low imaginary part of the complex refractive index (between 0.0012 and 0.003), and single scattering albedos more than 0.96 and without significant spectral dependence suggest that the analyzed particles present low absorption (and then low black carbon content) and a wide particle size distributions.

Integrated volume concentrations were obtained by assuming some reasonable features of the volume concentration profiles within the smoke layers, finding values of 0.016-0.038 $\mu m^3/\mu m^2$.





This integration was compared to the retrieved concentrations obtained with passive remote sensing retrievals, which usually provide information about the properties integrated along the whole atmospheric column. Particularly, a comparison was made with microphysical retrievals from three near AERONET stations. The similarity among the obtained intensive properties for the smoke layers and for the total atmospheric column is an indication that the tropospheric structure and properties were determined by the smoke plumes during those events.

As a practical application of the results, an approximately linear dependence was found between particle volume concentrations and extinction coefficients at 532 nm for the analyzed layers, and using also data from other studies. For the selected cases, this approximation is good and it can provide an estimation of the particle volume concentrations using only extinction when inversion algorithms cannot be applied. Nevertheless, it must be taken with caution, since these factors are only strictly applicable for similar aerosol particles (in terms of sources and aging) and vertical distributions.

**Acknowledgement**

This work was supported by the Andalusia Regional Government through project P12-RNM-2409, by the Spanish Ministry of Economy and Competitiveness through project CGL2013-45410-R and by the Spanish Ministry of Education, Culture and Sports through grant FPU14/03684. The financial support for EARLINET in the ACTRIS Research Infrastructure Project by the European Union's Horizon 2020 research and innovation program under grant agreement n. 654169 and previously under grant agreement n. 262254 in the 7[th] Framework Program (FP7/2007-2013) is gratefully acknowledged. The authors thankfully acknowledge the FEDER program for the instrumentation used in this work. This work was also partially funded by the University of Granada through the contract "Plan Propio. Programa 9. Convocatoria 2013" and through "Programa de Becas de Iniciación a la Investigación. Convocatoria 2014".

The Polish lidar development was financed by FNITP, Poland (Grant No.519/FNITP/115/2010). This work was supported by SONATA-BIS project funded by NCN, Poland (Grant No.2012/05/E/ST10/01578).

The authors express gratitude to the NOAA Air Resources Laboratory (ARL) for the HYSPLIT transport and dispersion model. We thank those at NRL-Monterey that have helped in the development of the NAAPS model and to the MODIS team for the use of FIRMS data. Acknowledgements are also due for NASA-EOS team members for providing CALIOP/CALIPSO datasets. We acknowledge those members of Granada, Leipzig and Belsk teams maintaining and supporting the instruments from AERONET network, and B. Holben and the AERONET team for the use of the retrievals and data availability.





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





| Station | Location | Lidar name and/or model | References |
|---|---|---|---|
| Atmospheric Physics Group, University of Granada, Spain (GR) | 37.16°N, 3.61°W, 680 m a.s.l. | MULHACEN, LR331-D400 | Guerrero-Rascado et al. (2008, 2009); Navas-Guzmán et al. (2013); Bravo-Aranda et al. (2013) |
| Leibniz Institute for Tropospheric Research, Leipzig, Germany (LE) | 51.35°N, 12.43°E, 90 m a.s.l. | Polly$^{XT}$ | Althausen et al. (2009); Engelmann et al. (2016); Baars et al. (2016) |
| Radiative Transfer Laboratory, University of Warsaw, Poland (WA) | 52.21°N, 21.03°E, 100 m a.s.l. | Polly$^{XT}$ | Althausen et al. (2009); Engelmann et al. (2016); Baars et al. (2016) |

**Table 1:** Geographical location of the selected EARLINET stations, and lidar system providing data for this study.

| | Layer Height a.s.l. (km) | $\beta_{532}$ (Mm$^{-1}$sr$^{-1}$) | $\alpha_{532}$ (Mm$^{-1}$) | $AE_{\alpha355\text{-}532}$ | $AE_{\beta355\text{-}532}$ | $LR_{355}$ (sr) | $LR_{532}$ (sr) | $CR_{LR}$ |
|---|---|---|---|---|---|---|---|---|
| GR | 5.20 ± 0.10 | 2.20 ± 0.09 | 82 ± 16 | 0.2 ± 1.2 | 1.2 ± 0.5 | 23 ± 10 | 51 ± 11 | 2 ± 1 |
| LE | 4.55 ± 0.15 | 0.93 ± 0.14 | 48 ± 5 | 0.3 ± 0.3 | 1.9 ± 0.4 | 25 ± 4 | 51 ± 9 | 2.1 ± 0.5 |
| WA | 2.28 ± 0.10 | 3.7 ± 0.5 | 216 ± 6 | 0.98 ± 0.06 | 1.9 ± 0.2 | 34 ± 6 | 58 ± 10 | 1.7 ± 0.4 |

5 **Table 2:** Average particle optical properties for the selected thin layers within the smoke zone.


Atmospheric Chemistry and Physics Discussions — Open Access



| Reference | Meas. type | Source region | Ageing | LR 355 (sr) | LR 532 (sr) | $AE_a$ (355/532) | $\delta_P$ | $CR_{LR}$ | RRI | IRI | $r_{eff}$ (µm) | SSA 355 nm | SSA 532 nm |
|---|---|---|---|---|---|---|---|---|---|---|---|---|---|
| Dubovik et. al (2002) | Sun Phot. | USA and Canada | All types | - | - | 1.0-2.3 | - | - | 1.50 ± 0.04 | 0.0094 ± 0.003 | - | 0.94 (440 nm) | 0.935 (670 nm) |
| Wandinger et al. (2002) | Lidar | NW Canada | 6-10 days-aged | - | 40-80 | - | - | - | 1.64-1.77 | 0.043-0.053 | 0.16-0.27 | - | 0.79-0.83 |
| | in-situ | NW Canada | 6-10 days-aged | - | - | - | - | - | - | - | 0.17-0.25 | - | 0.78-0.79 |
| Müller et al. (2005) | Lidar | Canada | 2 weeks-aged | 21-49 | 26-61 | 0.00-1.10 | - | - | 1.39-1.56 | 0.001-0.006 | 0.24-0.4 | - | - |
| Müller et al. (2005) | Lidar | Siberia | 3 weeks-aged | 21-67 | 31-87 | 0.27-1.10 | - | - | 1.37-1.6 | 0.001-0.007 | 0.24-0.38 | - | 0.89-0.98 |
| Müller et al. (2007a) | Lidar | Siberia/ Canada | aged | 46 ± 13 | 53 ± 11 | 1.0 ± 0.5 | < 5% | $(0.8 \pm 0.2)^{-1}$ | 1.49-1.53 | - | - | - | - |
| Alados-Arboledas et al. (2011) | Lidar | South Spain | 1 day-aged | 60-65 | 60-65 | 1.16-1.3 | - | 1 | 1.53 | 0.02 ± 0.02 | 0.13-0.17 | 0.76-0.83 | 0.80-0.87 |
| | Star Phot. | South Spain | 1 day-aged | - | - | 1.61 ± 0.10 | - | - | 1.61 | - | 0.19 ± 0.05 | - | - |
| Nicolae et al. (2013) | Lidar | Romania | Fresh | 43-73 | 43-46 | 1.37-1.93 | - | 0.6-1 | 1.61-1.66 | 0.009-0.05 | 0.27-0.4 | 0.74-0.92 | 0.74-0.94 |
| Nicolae et al. (2013) | Lidar | Greece | 2 days-aged | 41.1 ± 6.6 | 55.9 ± 7.8 | 1.28 ± 0.01 | - | 1.4 ± 0.5 | 1.65 ± 0.13 | 0.012 ± 0.08 | 0.34 ± 0.09 | 0.92 ± 0.07 | 0.87 ± 0.07 |
| Nicolae et al. (2013) | Lidar | Ukraine, Russia | 2-3 days-aged | 32-48 | 52-54 | 0.64-0.99 | - | 1.1-1.6 | 1.41-1.59 | 0.003-0.014 | 0.19-0.44 | 0.91-0.97 | 0.85-0.97 |
| Preißler et al. (2013) | Lidar | North America | 5-10 days-aged | 58 ± 17 | 56 ± 25 | 2.2 ± 0.7 | - | - | - | - | - | - | - |
| | Lidar | Iberian Peninsula | Fresh | 51 ± 17 | 54 ± 28 | 1.4 ± 0.5 | - | - | - | - | - | - | - |
| Pereira et al. (2014) | Lidar | Iberian Peninsula | 1-2 days-aged | 52-66 | 49-66 | 1.2-1.6 | 3.8-5% | 0.93-1.04 | 1.49-1.61 | 0.010-0.024 | 0.14-0.19 | 0.89-0.96 | 0.82-0.92 |
| Samaras et al. (2015) | Lidar | East Europe | aged | - | 27-55 | 1.2-2.3 | 4-8 % | - | 1.352-1.368 | $2.9 \cdot 10^{-4}$-0.0024 | 0.275-0.325 | - | 0.942-0.997 |
| Ancellet et al. (2016) | Satellite-based Lidar | North America | aged | 42-59 | 45-60 | - | 5-10% | - | - | - | - | - | - |
| Markowicz et al. (2016) | Sun Phot. | Canada | 5-6 days aged | - | - | 1.28-1.71 | - | - | - | - | - | 0.91-0.99 (441 nm) | - |

**Table 3:** optical and microphysical properties found in the literature about biomass burning events detected in Europe and used to compare with obtained values in Tables 2 and 4.





| | $C_v$ (μm³cm⁻³) | $r_{eff}$ (μm) | RRI | IRI | $SSA_{355}$ | $SSA_{532}$ |
|---|---|---|---|---|---|---|
| GR | 17.3 ± 0.2 | 0.33 ± 0.02 | 1.496 ± 0.017 | $(1.7 ± 0.4)·10^{-3}$ | 0.9820 ± 0.0002 | 0.9860 ± 0.0001 |
| LE | 10.1 ± 0.4 | 0.34 ± 0.03 | 1.480 ± 0.006 | $(3 ± 1)·10^{-3}$ | 0.965 ± 0.006 | 0.972 ± 0.004 |
| WA | 34.3 ± 0.7 | 0.207 ± 0.006 | 1.473 ± 0.002 | $(1.2 ± 0.3)·10^{-3}$ | 0.991 ± 0.001 | 0.99304 ± $5·10^{-5}$ |

**Table 4:** Average particle microphysical properties (namely volume concentration, effective radius, real and imaginary part of refractive index, and single scattering albedos) for the same selected thin layers within the smoke zone. The associate uncertainty for each variable corresponds to the standard deviation from the average solution.

| | $V_L$ (μm³ μm⁻²) | $V_L^f$ (μm³ μm⁻²) | $V_L^c$ (μm³ μm⁻²) |
|---|---|---|---|
| GR | 0.016 | 0.011 | 0.005 |
| LE | 0.016 | 0.01 | 0.006 |
| WA | 0.038 | 0.036 | 0.002 |

**Table 5:** Concentration values integrated along each smoke layer. Superscripts [f] and [c] indicate fine and coarse mode separation, respectively.





| | $r_{eff}$ (µm) | RRI | IRI | $SSA_{UV}$ | $SSA_{VIS}$ | V (µm³ µm⁻²) | $V^f$ (µm³ µm⁻²) | $V^c$ (µm³ µm⁻²) |
|---|---|---|---|---|---|---|---|---|
| GR: Cerro Poyos, 14.07.2013 06:29 UTC | 0.253 | 1.5044 | 0.013 | 0.9395 (438 nm) | 0.9325 (676 nm) | 0.037 | 0.031 | 0.007 |
| LE: 17.07.2013 17:31 UTC | 0.24 | 1.43 | 0.0005 | 0.9955 (441 nm) | 0.9951 (675 nm) | 0.072 | 0.051 | 0.021 |
| WA: Belsk, 09.07.2013 04:23 UTC | 0.23 | 1.52 | 0.014 | 0.9422 (439 nm) | 0.9214 (675 nm) | 0.067 | 0.049 | 0.018 |

**Table 6:** Columnar microphysical properties retrieved from AERONET inversions, which are the nearest in space and time to the analyzed lidar Raman measurements.