# Peer review of "Microphysical characterization of long-range transported biomass burning particles from North America at three EARLINET stations"

_Atmospheric Chemistry and Physics, 2016_

## Referee Comment (RC1) · Anonymous Referee #1 · 2 Jan 2017

This paper presents a comprehensive analysis of optical characteristics of transported aged forest fire smoke using multi-wavelength Raman lidars and AERONET sun photometers. The paper is very well written. The manuscript may be accepted in the present form. It would be better, however, to add some discussion on hygroscopic growth. Relationship between effective radius and relative humidity would be interesting. It would be also very interesting if vertical profile of effective radius in Warsaw was presented.

---

## Referee Comment (RC2) · Anonymous Referee #2 · 3 Jan 2017

General comment:

The paper discusses the microphysical properties of long-range transported biomass burning from N America within Europe, as determined from lidar measurements. The paper is in general clearly presented and the results properly discussed. The paper can be published after minor revisions.

Specific comments:

- pp 5, line 25 and fig 1; please define "smoke surface concentration"; is it PM?

[Figure]
* * *
Interactive
comment

- fig 3; please comment/explain why Warsaw data are not from the same smoke episode as for Granada and Leipzig; no measurements available? it would have been preferable to analyze the same smoke (i.e. having the same origin in time and space); also, there were no data available in Leipzig at the time of measurements in Warsaw; I would expect the smoke be seen both in Leipzig and Warsaw; was it a Calipso over-pass constraint?

- pp 11, line 10: please check LR for 532 for GR; according to Table 2, LR for GR should be ∼37 (82/2.2).

- pp 15, after lines 21; please comment on large differences for IRI between lidars and sun-photometers retrievals; Aeronet retrievals show larger IRI for GR and WA (∼90% difference with lidars) and much smaller IRI for LE (∼ 500% difference wrt lidars); different IRI are clearly reflected in different SSA; also, there are large difference in the concentration values as calculated from lidar and retrieved from sunphotometer

- pp 17, line 4: concerning the similarity for the intensive properties in the smoke layers... it looks to me that there is a good similarity for effective radius and RRI but not for IRI; please reconsider

---

## Referee Comment (RC3) · Anonymous Referee #3 · 11 Jan 2017

The authors have done an excellent job putting together models, back-trajectories and lidar data to retrieve microphysical properties of BB aerosols transported over the Atlantic ocean.

I am glad to see that my initial comments were taken into consideration, and I particularly like the new Table 3 with all the trans-Atlantic BB plume measurements in Europe. I agree with the publication of this manuscript after the authors address / reply to just a few minor comments below.

P.2 L.7-9: I've suggested the authors to give credit to the original papers about vertical

[Figure]

transport of BB aerosols, and I see that the authors simply included all the example articles that I mentioned. These are far too many. Please keep only those most relevant to your discussion.

P.6 L.20: In fig.3 I agree that one can see that the layer is somewhat decoupled from ground. However, since you looked at the LCL from the nearest radiosonde, and this is below the aerosol layer height, please say that too. It only makes your argument stronger.

P.7 L.5: The authors looked at their backtrajectory results in figure 4 (where one can see a few trajs that reach near ground) and argued that this is a proof "that the relevant air masses came from superficial layers (...)".

To my understanding, the fire power at ground level is strong enough to inject the BB at high altitudes. This is exactly why the atm-chem-models must have a plume-rise parametrization to calculate the injection height for each fire, otherwise they get the transport completely wrong.

If the authors have evidence that the smoke they observed should be comming from ground level (vegetation type? smoldering instead of flamming? something else?), they should present and discuss it.

P.8 L.11-13: Here the authors discuss figures 5, 6 and 7 which show that over the source (Canada and USA) the altitude of the smoke plume reaches up to 5km (see transects 1, 2, 8 and 9). This gives further confirmation of my previous comment. Over the source, you have smoke up to 5km, not just close to the ground.

Figure 5: Please mind color-blind or short-sighted readers, and change the green color to something that gives more contrast over the color used for the continents.

Figure 8: Please say (caption or text) how you selected which values to show. Are you masking out values when beta or alpha are lower than some threshold value? which?

[Figure]

[Figure]

---

## Author Comment (AC1) · 13 Mar 2017

The authors thank the reviewers for the efforts, time and the thorough review of our manuscript. Please, find below a detailed response to the reviewer's comments.

Comment: This paper presents a comprehensive analysis of optical characteristics of transported aged forest fire smoke using multi-wavelength Raman lidars and AERONET sun photometers. The paper is very well written. The manuscript may be accepted in the present form. It would be better, however, to add some discussion on hygroscopic growth. Relationship between effective radius and relative humidity would

be interesting. It would be also very interesting if vertical profile of effective radius in Warsaw was presented.

Response: We agree that a discussion on hygroscopic growth would be interesting, but we think that an analysis of those properties would deserve a more complete separated work if one wants to properly assess the enhancement factor and other related properties, which we consider that might be out of the scope of our paper. As suggested, we performed the calculation of the microphysical properties for several altitudes inside the detected smoke layer at Warsaw (namely $1.7\pm0.2$ km, $1.9\pm0.2$ km, and $2.1\pm0.2$ km) in order to retrieve vertical profiles of those properties. However, we found the same values (within uncertainties) as the ones retrieved for $2.28\pm0.2$ km, i.e., around $0.2~\mu$m for effective radii, around 1.47 for RRI and around 0.001 for IRI. For this reason, we decided to include in the manuscript only the retrieval corresponding to the highest values of particle backscatter and extinction coefficients, where the calculation appears more stable.

---

## Author Response (AR1)

The authors thank the reviewers for the efforts, time and the thorough review of our manuscript. Hereafter, the changes in the manuscript are noted here in italic and between quotation marks. The responses to the reviewer are marked in yellow on the manuscript.

**Anonymous Referee #1**

This paper presents a comprehensive analysis of optical characteristics of transported aged forest fire smoke using multi-wavelength Raman lidars and AERONET sun photometers. The paper is very well written. The manuscript may be accepted in the present form. It would be better, however, to add some discussion on hygroscopic growth. Relationship between effective radius and relative humidity would be interesting. It would be also very interesting if vertical profile of effective radius in Warsaw was presented.

**Answer:**

We agree that a discussion on hygroscopic growth would be interesting, but we think that an analysis of those properties would deserve a more complete separated work if one wants to properly assess the enhancement factor and other related properties, which we consider that might be out of the scope of our paper.

Following the reviewer's suggestions, we performed the calculation of the microphysical properties at additional altitudes inside the detected smoke layer at Warsaw (namely  $1.70\pm0.20$  km,  $1.90\pm0.20$  km, and  $2.10\pm0.20$  km) in order to retrieve vertical profiles of those properties. However, we found the same values (within uncertainties) as the ones retrieved for  $2.28\pm0.2$  km, i.e., around 0.2 µm for effective radii, around 1.47 for RRI and around 0.001 for IRI. For this reason, we decided to include in the manuscript only the retrieval corresponding to the highest values of particle backscatter and extinction coefficients, where where we got the best signal to noise ratio.

**Anonymous Referee #2**

General comment: The paper discusses the microphysical properties of long-range transported biomass burning from N America within Europe, as determined from lidar measurements. The paper is in general clearly presented and the results properly discussed. The paper can be published after minor revisions.

**Specific comments:**

- pp 5, line 25 and fig 1; please define "smoke surface concentration"; is it PM?

**Answer:**

The right panel in Fig. 1, provided by NAAPS (http://www.nrlmry.navy.mil/aerosol/), corresponds to a forecast of the smoke concentration at the surface level. According to Rubin et al. (Atmos. Chem. Phys., 16, 3927–3951, 2016), smoke emissions from biomass burning are

derived from satellite-based thermal anomaly data used to construct smoke source functions via the Fire Locating and Modeling of burning Emissions(FLAMBE) database. In order to make it clearer, we included a new sentence in the manuscript: "NAAPS (Navy Aerosol Analysis and Prediction System) model of Marine Meteorology Division, Naval Research Laboratory (NRL), (http://www.nrlmry.navy.mil/aerosol/) was used for forecasting aerosol optical depth and particle density of smoke at the Earth's surface, using smoke emissions derived from satellite-measured thermal anomalies." (pp 4, lines 16-19 in the new version)

- fig 3; please comment/explain why Warsaw data are not from the same smoke episode as for Granada and Leipzig; no measurements available? it would have been preferable to analyze the same smoke (i.e. having the same origin in time and space); also, there were no data available in Leipzig at the time of measurements in Warsaw; I would expect the smoke be seen both in Leipzig and Warsaw; was it a Calipso overpass constraint?

**Answer:**

During end of June and July 2013 several events of smoke transport were observed. The particle properties for those smoke events were varying, as the transport paths were not the same. Even for the cases at Leipzig and Granada, it is shown in the manuscript that the exact source regions may be different (either Canada or East USA). It is true that we employed the same CALIPSO overpasses to track the plumes for both stations (due to its availability), but we did not try to mean that the same event was detected.

We agree that an interesting idea would be to analyze the same smoke arriving at different stations, but the aim of this work is to characterize three different events of transatlantic smoke transport that happened within a certain time period, and that presented close sources and transport paths, in order to highlight similarities and differences among them.

About the data availability, we performed a search in the databases of each station and the presented cases were the existing measurements coinciding with smoke detection and that could be analyzed (unfortunately, not all the measurements can be analyzed because of cloud cover, signal instabilities, etc.).

Therefore, we welcome the suggestion, although we will not be able to include measurements of the same events measured at different stations.

**- pp 11, line 10: please check LR for 532 for GR; according to Table 2, LR for GR should be ~37 (82/2.2).**

**Answer:**

We checked the results of our optical profiles, and we found that indeed, the value  $LR_{532}$ = 51±11 sr for Granada was a misprint, and the right value is 47±11 sr. However, it still does not apparently coincide with the value obtained by directly dividing (82±16)/(2.20±0.09) = 37±9 sr from Table 2 (although they are not dramatically different taking into account uncertainties).

This fact is the result of the different smoothing and procedure applied: we obtained LR profiles (as in Fig. 8) from the ratio of  $\alpha$  and  $\beta$  profiles, each one retrieved with different smoothing as a consequence of the signals involved and the method used for retrieving each property; then, the average of the layer was taken from each individual profile, obtaining the values shown in Table 2. We think this is a more trustable procedure than directly dividing mean values, since we would then be involving different smoothings.

- pp 15, after lines 21; please comment on large differences for IRI between lidars and sunphotometers retrievals; Aeronet retrievals show larger IRI for GR and WA (~90% difference with lidars) and much smaller IRI for LE (~ 500% difference wrtlidars); different IRI are clearly reflected in different SSA; also, there are large difference in the concentration values as calculated from lidar and retrieved from sunphotometer

**Answer:**

It is true that the relative differences between lidar and sun-photometer retrievals are large, and it was not discussed in the text: we have now included a comment on it (as suggested) in the manuscript: "*Imaginary parts of refractive index values (IRI) showed larger differences with respect to values retrieved with lidar, what is also reflected in SSA. However, the SSA discrepancies remain less than 7% and then still represent low particle absorption.*" (pp 15, lines 21-23 new version).

It is still important to notice that the comparison cannot be very strictly done, considering several points: firstly, that the differences between ~10-3 (lidars) and ~10-2 or ~10-4 (photometers) do not imply too much difference in SSA (less than 7%), what means that in those ranges, the particle size distribution seems to play a more important role for the calculation of SSA than IRI; secondly, that according to Dubovik et al. (J. Geophys. Res., 111, 1984-2012, 2006), the uncertainties in IRI for small particles can be large. Additionally, one has to be careful when comparing particle microphysical properties retrieved for a certain altitude (as in lidar retrievals) and those retrieved for the whole atmospheric column (as AERONET retrievals), since the second retrievals include information about other aerosol layers in the atmospheric column not accounted in the lidar analysis at a specific layer. As commented in the manuscript, the fact that most of the properties are similar between lidar and AERONET retrievals mean that for the analyzed cases the columnar properties seem to be strongly influenced by the detected smoke layers, but it may not necessarily mean that all properties exactly coincide.

Concerning the concentration values, we would like to clarify that the values included in Table 4 (and named  $C_v$ ), in Tables 5 (named  $V_L$ ) and in Table 6 (named V) do not correspond to the same magnitude.  $C_v$  stands for the particle volume per unit air volume, and thus it is a magnitude defined for the single altitude we are investigating;  $V_L$  and V are height-integrated magnitudes, and thus refer to particle volume per unit air area, integrating only over the smoke layer to obtain  $V_L$ , or over the whole atmospheric column to obtain V. We included these different magnitudes in the tables in order to show the peak concentrations  $C_v$  (Table 4) and to assess the impact of the smoke layers (Table 5) on the whole column (Table 6). The percentages written in pp.15, lines 5-8, were indeed calculated dividing values in Table 5 over the ones in Table 6.

- pp 17, line 4: concerning the similarity for the intensive properties in the smoke layers... it looks to me that there is a good similarity for effective radius and RRI but not for IRI; please reconsider

**Answer:**

According to the answer to the previous comment, and with the sentences added in the corresponding section, the issue related to IRI similarity is also solved. We have included in the conclusions (page 17) that *"the majority"* of the properties are similar (referring to the discussion in the previous section).

**Anonymous Referee #3**

The authors have done an excellent job putting together models, back-trajectories and lidar data to retrieve microphysical properties of BB aerosols transported over the Atlantic ocean. I am glad to see that my initial comments were taken into consideration, and I particularly like the new Table 3 with all the trans-Atlantic BB plume measurements in Europe. I agree with the publication of this manuscript after the authors address / reply to just a few minor comments below.

P.2 L.7-9: I've suggested the authors to give credit to the original papers about vertical transport of BB aerosols, and I see that the authors simply included all the example articles that I mentioned. These are far too many. Please keep only those most relevant to your discussion.

**Answer:**

We gratefully accept the suggestion and keep just the papers more related to the information we want to demonstrate in the new version.

P.6 L.20: In fig.3 I agree that one can see that the layer is somewhat decoupled from ground. However, since you looked at the LCL from the nearest radiosonde, and this is below the aerosol layer height, please say that too. It only makes your argument stronger.

**Answer:**

In the new manuscript version, we include the reference to the nearest radiosonde at Legionowo in order to make our argument stronger, as suggested. We included "*This layer* was decoupled from the aerosol layer near to the surface, as it can be seen in Fig. 3, and was over the Lifted Condensation Level (LCL) according to nearest radiosonde at Legionowo (http://weather.uwyo.edu/upperair/sounding.html)." (pp 7, lines 1-2 new version)

P.7 L.5: The authors looked at their backtrajectory results in figure 4 (where one can see a few trajs that reach near ground) and argued that this is a proof "that the relevant air masses came from superficial layers (...)".

To my understanding, the fire power at ground level is strong enough to inject the BB at high altitudes. This is exactly why the atm-chem-models must have a plume-rise parametrization to calculate the injection height for each fire, otherwise they get the transport completely wrong.

If the authors have evidence that the smoke they observed should be comming from ground level (vegentation type? smoldering instead of flamming? something else?), they should present and discuss it.

**&**

P.8 L.11-13: Here the authors discuss figures 5, 6 and 7 which show that over the source (Canada and USA) the altitude of the smoke plume reaches up to 5km (see transects 1, 2, 8 and 9). This gives further confirmation of my previous comment. Over the source, you have smoke up to 5km, not just close to the ground.

**Answer:**

We fully agree with your considerations about smoke plumes injection height, and thus we changed the word "superficial" in the manuscript to avoid misunderstanding.

Figure 5: Please mind color-blind or short-sighted readers, and change the green color to something that gives more contrast over the color used for the continents.

**Answer:**

We changed the green color to purple.

Figure 8: Please say (caption or text) how you selected which values to show. Are you masking out values when beta or alpha are lower than some threshold value? which?

**Answer:**

In text (pp. 10, lines11-12) we say that "The regions of profiles affected by incomplete overlap and by too low backscatter ratio are not shown". In particular, we avoided regions with backscatter coefficient less than  $0.2 \cdot 10^{-3}$  km-1sr-1.

**List of all relevant changes made in the manuscript (pages and lines referred to the new uploaded manuscript version):**

page 2, lines 7-8: references updated.

**page 4, lines 16-19:** statement "*using smoke emissions derived from satellite-measured thermal anomalies*" added to the original sentence.

**page 7, lines 1-2:** we included "This layer was decoupled from the aerosol layer near to the surface, as it can be seen in Fig. 3, and was over the Lifted Condensation Level (LCL) according to nearest radiosonde at Legionowo (http://weather.uwyo.edu/upperair/sounding.html)."

page 7, line 8: "superficial layers" changed by "areas over North American forest fires".

page 8, figure 5 and page 9, figure 7: green colour changed to purple.

page 11, line 9; page 16, line 14 and page 25, table2: "51" replaced by "47" (misprint).

**page 15, lines 21-23:** sentence "Imaginary parts of refractive index values (IRI) showed larger differences with respect to values retrieved with lidar, what is also reflected in SSA. However, the SSA discrepancies remain less than 7% and then still represent low particle absorption." added.

page 17, line 4: "The majority of" included.

**Microphysical characterization of long-range transported biomass burning particles from North America at three EARLINET stations**

Pablo Ortiz-Amezcua1,2, Juan Luis Guerrero-Rascado1,2, María José Granados-Muñoz1,3, José Antonio Benavent-Oltra1,2, Christine Böckmann4, Stefanos Samaras4, Iwona S. Stachlewska5, Łucja Janicka5, Holger Baars6, Stephanie Bohlmann6 and Lucas Alados-Arboledas1,2

[revised manuscript text omitted]